# Integrating high resolution drone imagery and forest inventory to distinguish canopy and understory trees and quantify their contributions to forest structure and dynamics

**Raquel Fernandes Araujo**[1,2]*, **Jeffrey Q. Chambers**[3], **Carlos Henrique Souza Celes**[1], **Helene C. Muller-Landau**[2], **Ana Paula Ferreira dos Santos**[1], **Fabiano Emmert**[4], **Gabriel H. P. M. Ribeiro**[1,5], **Bruno Oliva Gimenez**[1,2], **Adriano J. N. Lima**[1], **Moacir A. A. Campos**[1], **Niro Higuchi**[1]

1 Laboratório de Manejo Florestal, Instituto Nacional de Pesquisas da Amazônia, Manaus, Amazonas, Brazil, 2 Center for Tropical Forest Science, Smithsonian Tropical Research Institute, Gamboa, Panama, Republic of Panama, 3 Geography Department, University of California, Berkeley, California, United States of America, 4 Instituto de Ciências Agrárias, Universidade Federal Rural da Amazônia, Belém, Pará, Brazil, 5 Faculdade de Engenharia Florestal, Universidade Federal de Mato Grosso, Cuiabá, Mato Grosso, Brazil

* araujo.raquelf@gmail.com

## Abstract

Tree growth and survival differ strongly between canopy trees (those directly exposed to overhead light), and understory trees. However, the structural complexity of many tropical forests makes it difficult to determine canopy positions. The integration of remote sensing and ground-based data enables this determination and measurements of how canopy and understory trees differ in structure and dynamics. Here we analyzed 2 cm resolution RGB imagery collected by a Remotely Piloted Aircraft System (RPAS), also known as drone, together with two decades of bi-annual tree censuses for 2 ha of old growth forest in the Central Amazon. We delineated all crowns visible in the imagery and linked each crown to a tagged stem through field work. Canopy trees constituted 40% of the 1244 inventoried trees with diameter at breast height (DBH) > 10 cm, and accounted for ~70% of aboveground carbon stocks and wood productivity. The probability of being in the canopy increased logistically with tree diameter, passing through 50% at 23.5 cm DBH. Diameter growth was on average twice as large in canopy trees as in understory trees. Growth rates were unrelated to diameter in canopy trees and positively related to diameter in understory trees, consistent with the idea that light availability increases with diameter in the understory but not the canopy. The whole stand size distribution was best fit by a Weibull distribution, whereas the separate size distributions of understory trees or canopy trees > 25 cm DBH were equally well fit by exponential and Weibull distributions, consistent with mechanistic forest models. The identification and field mapping of crowns seen in a high resolution orthomosaic revealed new patterns in the structure and dynamics of trees of canopy vs. understory at this site, demonstrating the value of traditional tree censuses with drone remote sensing.

**Data Availability Statement:** All relevant data are within the paper and its Supporting information files.

**Funding:** This study was financed by the INCT - Amazonian Woods (FAPEAM/CNPq) and by the Next Generation Ecosystem Experiments-Tropics, funded by the U.S. Department of Energy, Office of Science, Office of Biological and Environmental Research under Contract DE-AC02-05CH11231. Fellowship support for RFA was provided from Coordination for the Improvement of Higher Education Personnel (CAPES). The funders had no role in study design, data collection and analysis, decision to publish, or preparation of the manuscript.

**Competing interests:** The authors have declared that no competing interests exist.

# Introduction

Scientists have long sought to understand tropical forest structure and dynamics—the abundances of trees of different sizes and canopy positions, and their growth and mortality rates [1]. Whether trees are in the canopy (i.e., directly exposed to overhead light) or not has long been recognized as a critical determinant of tree performance [2–6]. However, the tall canopy height and dense understory of many tropical forests often make it difficult to establish whether an individual tree is in the canopy or in the understory, because it is hard to see the tops of crowns and assess their light exposure from the ground. Cameras mounted on drones, technically called remotely piloted aircraft systems (RPAS), produce high spatial resolution images with pixel size < 5cm that enable the visualization of individual tree crowns [7–12]. Thus, the integration of RPAS remote sensing and ground-based data provides the opportunity for the exact determination of canopy status to be linked with information on tree diameter, growth, etc., thereby enabling new insights into the structure and dynamics of tropical forests.

Remote sensing provides increasing amounts of information about tropical forests including phenology, photosynthesis, and functional composition, but the signal in many types of data is largely determined by canopy trees [13–16]. In contrast, ground-based plot data include both canopy and understory trees. Thus, a key issue in linking and integrating ground-based and remote sensing datasets is understanding which trees are in the canopy, and their role in the forest. In particular, what proportion of trees of different sizes are in the canopy, and what are their contribution to forest carbon stocks and woody productivity? It is well-known that larger trees are more likely to be in the canopy, that larger trees contribute disproportionately to forest carbon stocks, and that woody productivity increases with tree size and light exposure [17–25]. But few studies have quantified how canopy position varies with tree size and growth rates or how canopy trees contribute to growth rates and carbon stocks and fluxes. A rare exception is work combining airborne imagery and ground-based data for the old-growth moist tropical forest of Barro Colorado Island, Panama, to evaluate the canopy status of individual trees, quantify the proportion of trees in the canopy, and compare diameter growth of canopy and understory trees [26, 27].

Forest carbon stocks and productivity are closely related with tree size distributions, a fundamental attribute of forest structure [28–30]. Whole-forest tree size distributions can be understood as the sum of size distributions of understory and canopy trees, which are shaped by different processes [31]. However, to date, tests of theories explaining tree size distributions have been conducted exclusively at the level of the whole stand, without distinguishing between canopy and understory trees [32–36]. Metabolic ecology derives a power function tree size distribution from arguments regarding the scaling of metabolic rates with diameter, and specifically predicts that the diameter distribution follows a power function with exponent -2, i.e., $N \sim D^{-2}$, for N trees of D diameter [32, 33]. Demographic equilibrium derives tree size distributions from the von Foerster equation and empirical relationships for growth and mortality with size, and predicts that diameter distributions will be better fit by Weibull and Quasi-Weibull functions [34–36]. In contrast, the more mechanistic approach of Farrior et al. [31] predicts that canopy trees will follow an exponential distribution whereas understory trees will follow a power function. The approach of Farrior et al. [31] parallels the structure of vegetation demographic models, taking into account multiple size-classes and light environments [37].

In this study, we use the combination of images collected with digital camera mounted on RPAS and detailed field mapping of tree crowns to determine the canopy status of individual trees and link this information to forest inventory data in an old-growth forest near Manaus, Brazil in the Central Amazon. We thus determine the proportions, growth rates and size distributions of canopy and understory trees, as well as the contributions of canopy and understory

trees to total biomass and wood productivity. We specifically addressed the following questions: (i) What is the proportion of trees in the canopy and how does this vary with diameter? (ii) How do growth rates differ between canopy and understory trees, and how does that difference vary with diameter? (iii) What are the relative contributions of canopy and understory trees to aboveground biomass carbon stocks and aboveground wood productivity? (iv) What are the forms of size distributions of canopy trees and understory trees, how do they differ from each other and from whole-forest size distributions, and how do they fit with competing theories?

## Materials and methods

### Study site

The study was carried out in the northernmost 1020 m of a North-South Transect Plot that is located in the Estação Experimental de Silvicultura Tropical (EEST ZF-2) of the Instituto Nacional de Pesquisas da Amazônia (INPA) (Fig 1), a research reserve with 21,000 ha. The area is covered by old-growth terra-firme forest, characterized by a closed canopy with high tree species diversity and a dense understory [38, 39]. The North-South and East-West

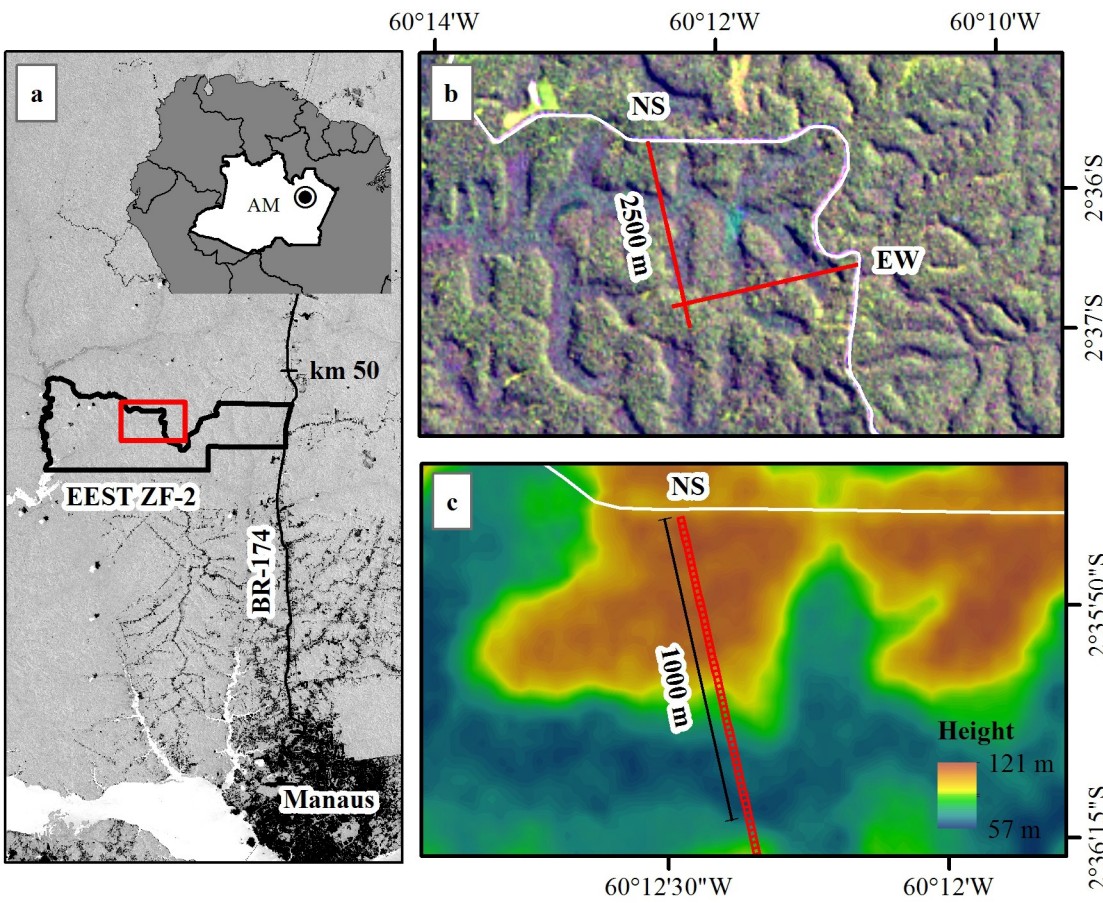

**Fig 1. Map of the study area.** (a) The EEST-ZF-2 study site is located 50 km north of Manaus, Brazil. (b) The North-South and East-West Transect Plots are permanent plots of 20 x 2500 m each. (c) The RPAS overflight covered the first 1020 m of the NS plot, which includes representation of plateau, slope and valley areas. Landsat-8 (a, b) and SRTM (c) images courtesy of the U.S. Geological Survey.

Transect Plots are permanent inventory plots that were installed in 1996 by the Jacaranda Project (a collaboration between INPA and Japan International Cooperation Agency, JICA) with dimensions of 20 x 2500 m each, totaling 10 ha. These transect plots were designed to representatively sample the dominant undulating topography of the region, which encompasses plateau, slope and valley positions and associated forest structural and functional differences (Fig 1b). The transects were subdivided into 20 m x 20 m subplots (125 for each transect) and the forest dynamics (growth, recruitment, mortality) was monitored with repeat censuses of trees with DBH > 10 cm (S1 File). Censuses were performed in 1996, 2000, 2002, 2004, 2006, 2008, 2010, 2011, 2013 and 2015, for a total of 10 inventories between 1996 and 2015, inclusive (S2 and S3 Files). The availability of ~two decades of bi-annual tree censuses and the proximity from the road (Fig 1) enabled the accomplishment of the present study in this plot.

## Image acquisition and processing

Digital RGB camera imaging with an RPAS was performed on the northernmost 1020 m of the North-South transect, covering 51 20 m x 20 m subplots, in April 2016 (Fig 1c). We employed a DJI Phantom 2 with an RGB camera mounted on a three-axis gimbal. We replaced the standard GoPro lens with a lens having a 5.5 mm focal length and 60˚ Field of View (FOV). The resulting photos have a resolution of 12 Mp with dimensions of 4000 x 3000 pixels. The flight was made at 80 m above ground, speed of 4 $m.s^{-1}$ and spacing between flight lines of 10 m. Photos were taken every 1 second, covering an area of ~60 m wide at the height of the canopy. The minimum longitudinal and side overlap were 88% and 78%, respectively, in areas of peak canopy height on ridges; overlap was larger in areas with lower canopy height and in slope and valley areas.

We processed photos using the photogrammetry software Agisoft Photoscan (https://www.agisoft.com, v.1.3.0), which aligned the photos using the Scale Invariant Feature Transformation (SIFT) algorithm [40] and produced a point cloud model based on overlap among photos (because the camera was not integrated with the RPAS, GPS coordinates of the flight were not automatically assigned to photos). We georeferenced this model using as reference an Airborne Laser Scanning (ALS) dataset from the North-South Transect plot. We selected 15 control points evenly distributed across the flight area and extracted the XYZ coordinates (UTM —Universal Transverse Mercator projection Zone 20S, WGS84 horizontal datum). The control points were the center of crowns and palm trees, and the solar panels of two towers of the AmazonFACE project located in the plot, all well visible in both ALS data and photos. Georeferencing accuracy was assessed in terms of the Root Mean Square Error (RMSE) reported by the software. We then generated a 3D point cloud, a digital elevation model (DEM) and a 2 cm spatial resolution orthomosaic (e.g., Fig 2; S4 File) using Agisoft Photoscan.

## Crown delineation and linkage to tagged trees

Crowns visible in the orthomosaic were associated with tagged trees through field work, enabling us to link almost two decades of bi-annual forest dynamics data with detailed crown characteristics for the first time at this site. Maps with the orthomosaic were printed for each 20 m x 20 m subplot and crowns visible in the image were identified in the field, with reference to the forest inventory data on individual tree DBH, species identity, and tag number. We started delineating the crowns of trees with the biggest trunk diameters, looking from the base to the top and identifying the whole boundary of each crown. We drew the tree crowns boundaries on the map with their respective tag numbers. We then delineated the crowns of the smaller trees, identifying the neighboring trees which had crowns surrounding the biggest crowns. This *in situ* delineation of crowns allowed us to perceive that in some cases branches

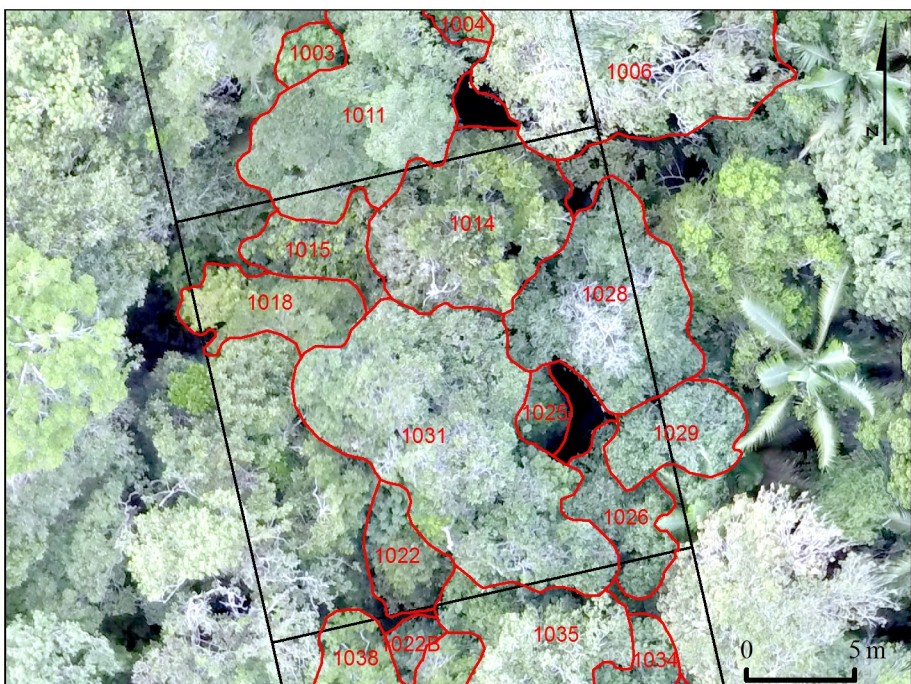

**Fig 2. Orthomosaic image showing canopy tree crowns mapped in the field.** Numbers in red are the assigned tree tags; the black lines correspond to the limits of each 20 m x 20 m subplots.

of the same tree, but located in different heights and positions, had different colors in the orthomosaic, such that examination of the imagery alone would suggest they belong to different individuals. After the field work, all the crowns were vectorized in GIS, with the creation of polygons to represent each crown (Fig 2; S5 File). We used the orthomosaic as the reference and vectorized the polygons using the same projected coordinate system. Finally, we returned to the field to address some questions that arose during the vectorization of crowns, and thereby minimize errors in the linkages of delineated crowns with forest inventory data.

## Data analysis

**Defining canopy status.** For the purposes of this study, canopy trees were defined as those directly exposed to overhead light; other trees were classified as understory trees (as in Bohlman [27]). Specifically, trees were classified as being in the canopy if their crowns were totally or partially visible in the orthomosaic, and had a visible crown diameter greater than ~ 1.5 m. We note that some trees classified as canopy trees under this definition may have most of their crowns shaded and only a small part of the crown in direct light, and that small trees can be classed as canopy trees if they do not have larger trees above them. We also note that some trees classified as understory trees under this definition may received direct lateral light for part of the day.

**Canopy status by size.** We calculated the proportions of trees that were in the canopy (exposed) or understory (non-exposed) for each 10 cm wide diameter class, and for all trees combined. Confidence intervals on these proportions were obtained from 1000 bootstraps over the 20 m x 20 m subplots. To quantify size-dependence of canopy status, we conducted a

logistic regression of canopy status against stem diameter (Eq 1):

$$f(x) = \frac{1}{1 + e^{-(a+bx)}} \tag{1}$$

where $\alpha$ and $b$ are parameters, $x$ is the independent variable DBH in cm, and $e$ is natural exponential basis.

**Growth.** We calculated the mean annual increment (MAI) of each tree from DBH measurements taken from 1996 to 2015 (1996, 2000, 2002, 2004, 2006, 2008, 2010, 2011, 2013, 2015). Specifically, we calculated the mean growth rate of each tree as the slope (b) of the regression of DBH against time. We included only trees that had at least four DBH measurements (i.e., trees that recruited in 2010 or earlier). We compared MAI distributions between canopy and understory trees using the Levene variance test and the Student's t-test. We conducted log-log regressions of MAI against DBH, fitting separate functions for canopy and understory trees. Because negative and zero growth values cannot be included in log-log regressions, we replaced all negative MAI (16 trees) with half of the smallest positive MAI value for this analysis.

**Predicting canopy status.** We fitted logistic regressions for canopy status as a function of DBH alone, MAI alone, and DBH and MAI combined. We evaluated the classification accuracy of canopy/understory status based on the fitted functions (i.e., classifying trees as canopy or understory based on whether the predicted canopy probability was greater or smaller than 0.5, respectively) by calculating the Kappa concordance index and the global accuracy derived from the confusion matrix [41].

**Contribution of canopy trees to carbon and above ground wood productivity.** We estimated the above ground carbon stock of each tree using the biomass equation and water and carbon contents of Silva [42]:

$$AGB = 2.2737 DBH^{1.9156} (n = 494, R^2 = 0.85, \text{uncertainty} = 8.4\%) \tag{2}$$

$$AGC = AGB(1 - WC)CC \tag{3}$$

where $AGB$ is the fresh above ground biomass in kg, $DBH$ is the diameter at breast height in cm; $AGC$ is the above ground carbon in kg, $WC$ is the water content (here 0.408) and $CC$ is the carbon content (here 0.485) [43]. We then calculated the proportional contribution of canopy trees to total estimated aboveground carbon.

We calculated the wood productivity (kg C.yr$^{-1}$) of each tree and of the stand as a whole, and the proportional contribution of canopy trees, using two approaches. In the first approach, we calculated growth for 2011–2015 using observed DBHs for those years. In the second approach, we estimated DBH in 2011 and 2015 using the equation fit to the entire DBH time series from 1996 to 2015, thus effectively using average growth over that entire time period. In both cases, the observed or estimated DBHs in 2011 and 2015 were combined with Eqs 2 and 3 to estimate AGC on both dates, and their difference was used to calculate woody productivity in kg C.yr$^{-1}$. Note that the first approach has the advantage that growth is closer in time to the canopy status measurements, but the disadvantage that individual measurement errors have more influence (in particular, some trees exhibited negative productivity).

**Tree size distributions.** We quantified the size distributions for all trees, canopy trees, and understory trees. In each case, we fit three alternative probability distributions—

exponential, power, and Weibull (Eqs 4–6)–using maximum likelihood [44, 45]:

$$f(x) = \lambda e^{-\lambda x} \tag{4}$$

$$f(x) = x^{-\lambda} \tag{5}$$

$$f(x) = \frac{k}{\lambda}\left(\frac{x}{\lambda}\right)^{k-1} e^{-\left(\frac{x}{\lambda}\right)^k} \tag{6}$$

where $\lambda$ and $k$ are fitted parameters, $x$ is DBH in cm, and $e$ is the natural exponential basis. Trees were first binned in 1 cm classes from Dmin = 11 cm to Dmax = 117 cm (the 10–11 cm size class was omitted from analysis because of inconsistencies in measurements associated with the lower size cutoff at 10 cm). For maximum likelihood estimation, the PDFs were normalized to sum to one over the focal diameter range, 11–117 cm. The maximum likelihood estimates of the parameters were those that maximized the likelihood function (Eq 7):

$$L = \sum_i N_i \log[F(x_{max,i}) - F(x_{min,i})] \tag{7}$$

where the summation is over the size class intervals $i$, $N_i$ is the number of trees in size class $i$, $F(x)$ is the cumulative probability distribution of $f(x)$, $x_{max,i}$ and $x_{min,i}$ are the minimum and maximum DBH in size class $i$, and thus the quantity in square brackets is the total probability an individual is in size class $i$ under the candidate parameters and probability density function. We used Akaike's Information Criterion (AIC) to compare the goodness of fits of the different functions [46]. We obtained 95% confidence intervals on parameters from 1000 bootstraps over the 20 x 20 m subplots.

## Results

Of the 1244 trees with DBH > 10 cm in the first 51 subplots of the NS Transect Plot, 40% (498 trees) were in the canopy, with at least parts of their crowns exposed and visible in the image. This represents a density of 249 canopy trees per ha. The proportion of trees in the canopy increased with diameter from 21% for trees 10–20 cm to 57% for 20–30 cm, up to 100% for trees above 70 cm (Fig 3, S1 Table). Logistic regression provided a reasonably good fit to the proportion of trees in the canopy (Fig 3). The fitted equation predicts that a tree 23.5 cm DBH has 50% probability of being in the canopy (Fig 3).

Growth of canopy trees averaged 2.34 mm.year$^{-1}$ (CI 0.18 mm.year$^{-1}$), just twice that of understory trees, which averaged 1.18 mm.year$^{-1}$ (CI 0.07 mm.year$^{-1}$; n = 484 and 696 trees, respectively; S1 Fig). Canopy and understory trees differed significantly in mean growth rates (t-test, p<0.001), and in the variances of growth rates (Levene variance test, p<0.001). Growth rates were approximately a power function of diameter for understory trees, and were not significantly related to diameter in canopy trees (Fig 4a and 4b). At small diameters, canopy trees had much higher growth rates than understory trees; this difference decreased with increasing diameter (Fig 4c).

The probability of a tree being in the canopy was reasonably well-predicted from DBH, and somewhat better predicted using DBH and prior growth (MAI) in combination (Table 1). Growth alone was not as good a predictor as DBH. The fitted logistic regression based on MAI alone crossed 50% probability of canopy status at 2.2 mm.year$^{-1}$ (S2 Fig).

Canopy trees accounted for a disproportionately large share of carbon stocks and fluxes. Though canopy trees were only 40% of all trees greater than 10 cm DBH, they accounted for 67% of the total above ground carbon stocks. In terms of wood productivity, we estimated that

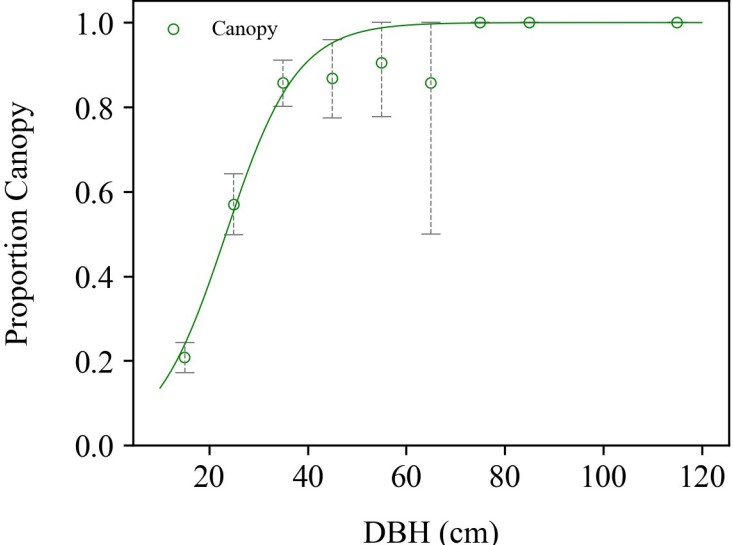

**Fig 3. Proportions of trees in the canopy.** Observed proportions of trees in the canopy in each 10 cm DBH class (green points), together with the fitted logistic regression (solid green line). Dashed vertical bars give 95% confidence intervals from bootstrapping over subplots.

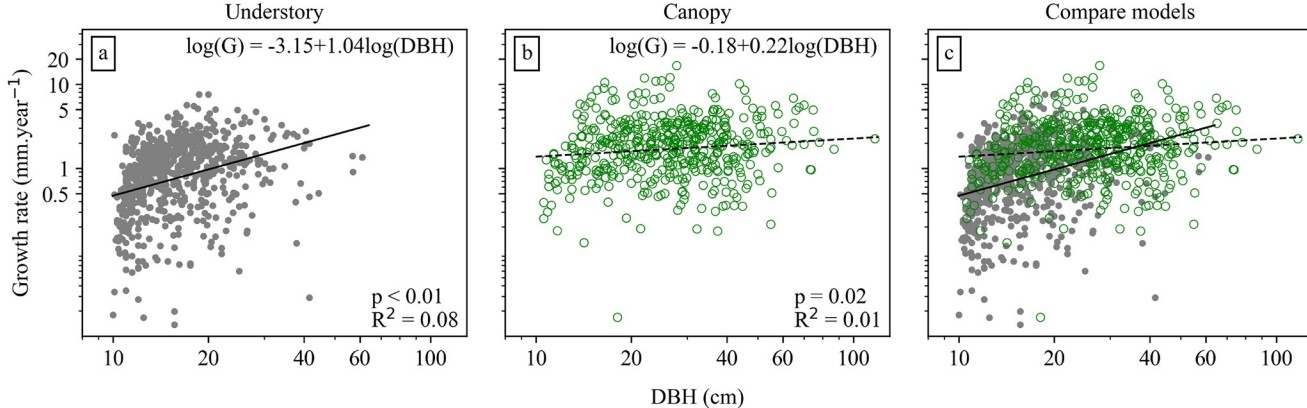

**Fig 4. Relationship of diameter growth with DBH for understory and canopy trees.** Fitted lines are linear regressions of log-transformed data, with solid lines indicating that the slope is significantly different from zero, and dashed lines that it is not.

**Table 1. Model coefficients and fit statistics for logistic regressions and associated classifications of canopy status of individual trees based on DBH, growth (MAI) or both combined.**

|  | DBH only | Growth only | DBH and Growth |
|---|---|---|---|
| Intercept ($a$) | -3.276 | -1.413 | -3.723 |
| DBH ($b_1$) | 0.140 | NA | 0.124 |
| Growth ($b_2$) | NA | 0.658 | 0.457 |
| Overall accuracy | 0.764 | 0.681 | 0.790 |
| Kappa | 0.492 | 0.299 | 0.553 |

canopy trees contributed 75% when using growth data for the last 4 years alone, and 68% of wood productivity when using growth data for the entire previous 19-year period.

The size distribution of canopy trees differed from those of understory trees, and both distributions differed from those of all trees combined (Fig 5; S4 Fig). Among the tested models, the best fit for all individuals was the Weibull distribution (Fig 5a and 5c; Table 2). The best fit for the understory trees was the exponential distribution, with the Weibull producing an almost equally good fit (Fig 5a and 5c; Table 2; S4 Fig). The canopy tree size distribution was unimodal (Fig 5b and 5d) and poorly fit by the exponential distribution (S3 Fig). Because trees with DBH $\geq$ 25 cm had more than 50% probability of being in the canopy (Fig 3), we took a DBH of 25 cm as a logical threshold diameter for separately fitting canopy trees. The size distribution of canopy trees with DBH $\geq$ 25 cm was best fit by the exponential function, with the Weibull producing a similarly good fit (Fig 5b and 5d; Table 2; S4 Fig).

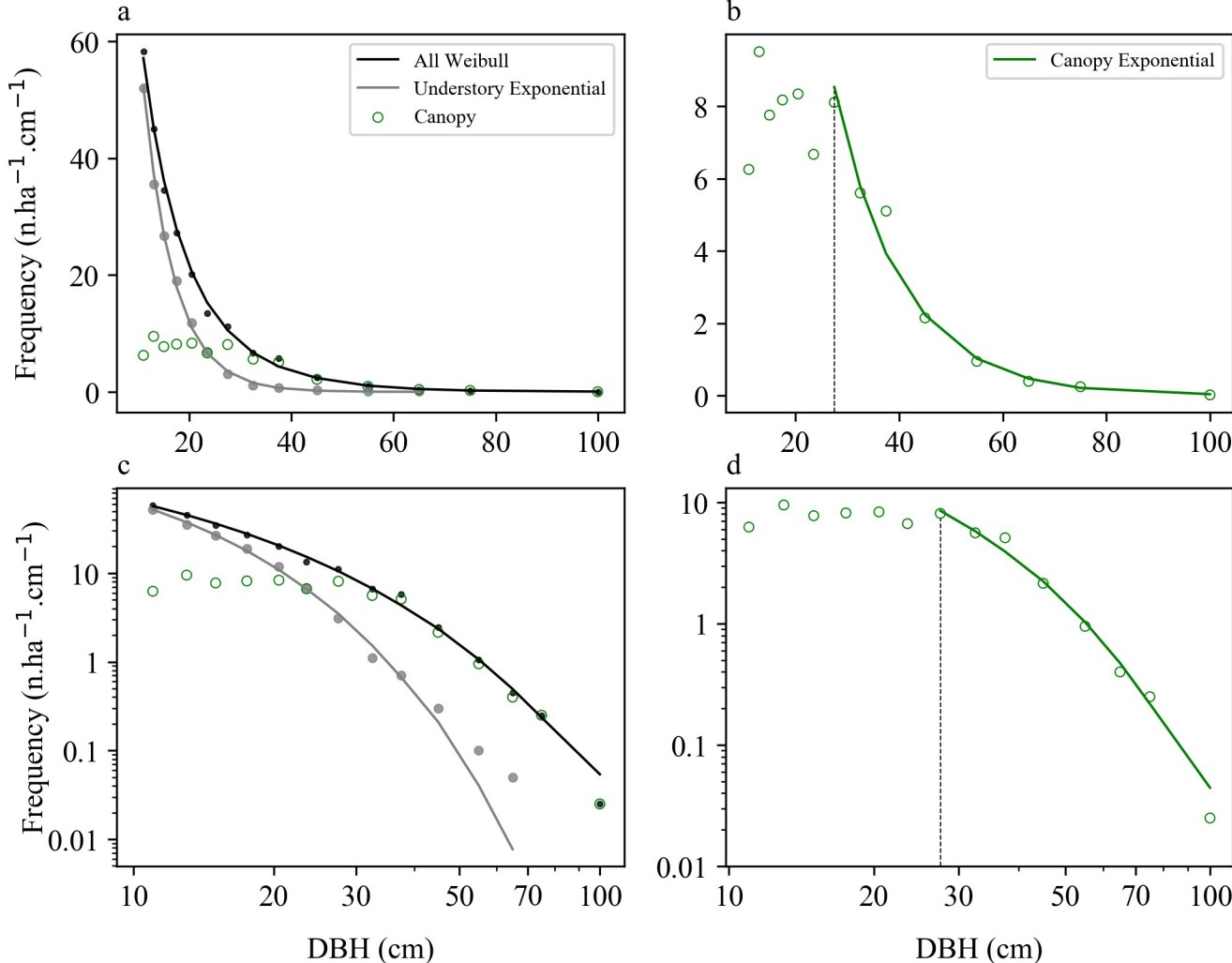

**Fig 5. Observed size distributions of understory trees, canopy trees, and both combined.** Size distributions of all 1244 trees with DBH > 10 cm (black points), understory trees (gray points), and canopy trees (green circles), shown together with best-fit probability density functions (lines). Distributions are shown on both linear (top) and log (bottom) scales. The vertical dashed black line indicates the minimum diameter (25 cm) for inclusion in the canopy tree fits. The Weibull distribution (Eq 6) had the best fit for all individuals combined; the exponential distribution (Eq 4) had the best fit for understory trees as well as for canopy trees with DBH $\geq$ 25 cm. Whereas data are graphed here for 10-cm size classes for visualization purposes, fits were carried out using 1-cm size classes (parameter values in Table 2).

**Table 2. Parameter values and delta AIC values for maximum likelihood fits of exponential, power and Weibull probability density functions to size distributions for all trees, understory trees, canopy trees, and canopy trees ≥ 25 cm DBH.**

| Group | Distribution | λ (95% CI) | k (95% CI) | Delta AIC |
|---|---|---|---|---|
| All | Exponential | 0.091 (0.085–0.097) | | 5.50 |
| All | Power | 2.539 (2.438–2.639) | | 77.26 |
| **All** | **Weibull** | **7.299 (4.914–9.762)** | **0.805 (0.691–0.925)** | **0.00** |
| **Understory** | **Exponential** | **0.165 (0.154–0.180)** | | **0.00** |
| Understory | Power | 3.558 (3.410–3.741) | | 37.82 |
| **Understory** | **Weibull** | **4.206 (2.500–7.349)** | **0.851 (0.704–1.128)** | **0.54** |
| Canopy | Exponential | 0.056 (0.051–0.061) | | 39.49 |
| Canopy | Power | 1.734 (1.623–1.856) | | 167.64 |
| **Canopy** | **Weibull** | **27.082 (24.241–29.531)** | **1.641 (1.416–1.880)** | **0.00** |
| **Canopy≥25** | **Exponential** | **0.078 (0.070–0.086)** | | **0.00** |
| Canopy≥25 | Power | 3.451 (3.197–3.745) | | 18.91 |
| **Canopy≥25** | **Weibull** | **17.531 (5.528–27.771)** | **1.19 (0.716–1.796)** | **1.28** |

Delta AIC is the difference in AIC from the best model. The best-fit models for each dataset, and those within 2 delta AIC of the best model, are highlighted in bold.

## Discussion

Our integration of high resolution drone imagery and forest inventory data enabled us to define canopy positions of individual trees, and quantify structural and dynamic contributions of canopy and understory trees in this old-growth tropical forest in Central Amazonia. We found that canopy trees constituted 40% of trees with DBH > 10 cm and accounted for ~70% of carbon stocks and wood productivity. Diameter growth of canopy trees was on average twice as large as that of understory trees, and was size-independent in canopy trees. The size distribution of canopy trees differed markedly from that of understory trees and from the whole-forest size distribution; distributions of understory trees and of canopy trees with DBH ≥ 25 cm were well fit by the exponential functions, whereas the whole-forest distribution was much better fit by a Weibull. These findings contributed to improve the understanding of the structure and dynamics of trees with similar light environments in tropical forests.

The probability of being in the canopy increased logistically with tree diameter, passing through 50% at 23.5 cm DBH. Overall there were 249 trees.ha$^{-1}$ in the canopy, 40% of trees > 10 cm DBH at our study site. We know of only one comparable study, of old-growth moist tropical forest on Barro Colorado Island (BCI), Panama, where there were 215 trees.ha$^{-1}$ in the canopy in 2015, constituting 50% of trees > 10 cm DBH, and where the probability of being in the canopy reached 50% at less than 20 cm DBH [26, 27]. Our site also has a substantially higher density of trees overall, with 622 trees > 10 cm per hectare, compared with 430 for BCI [47]. Thus, our site has a somewhat higher density of canopy trees, and a much higher density of understory trees (373 vs. 215 trees.ha$^{-1}$), consistent with the presence of trees of larger sizes in BCI [48]. The observed proportions of trees in the canopy for different tree size classes at BCI matched those predicted by the perfect plasticity approximation (PPA) when run at the scale of 31.25 m width subplots [26]. The PPA algorithm "fills" the canopy layer crown by crown starting from the tallest trees (the canopy is full when the summed crown area of canopy trees equals or exceeds the area of the relevant subplot). Investigating the proportion of trees in the canopy per species or genus should bring more information on forest structure heterogeneity. At BCI, the proportion of gap-dependent species was higher in the canopy than the understory and increased with tree size, while the proportion of shade-tolerant species was smaller

in the canopy and decreased with tree size [27]. Future studies should evaluate if the PPA can explain differences in the numbers and sizes of canopy trees among forests such as those between our site and BCI and explore differences in species and functional group compositions between canopy and understory. Expanding this approach for larger areas and other forest types would increase the understanding of the general proportion of trees occupying the canopy in relation to tree size, contributing to predictions of forest structure from satellite imagery.

Light availability is one factor that directly influences tree growth in tropical forests. In this study we found that growth of canopy trees was on average twice as large as that of understory trees. Similarly, a previous study of BCI found that diameter increment was 2.6 times larger in canopy trees than in understory trees [26]. Other studies showed a positive relationship of diameter growth with crown exposure as assessed by ground-based observers [17, 19–21]. We found that canopy trees did not show an increase in growth with diameter, consistent with all of them having high light exposure regardless of size. The positive relationship of growth with diameter in understory trees suggests that light availability increases with size within the understory. This implies that the general increase in growth with diameter across all trees combined is due to increasing average light with diameter [35].

Higher light availability and thus higher growth rates for canopy trees translate into a high proportion of stand-level woody productivity, here an estimated 68–75%. The higher estimate of wood productivity based on growth in the most recent 4 years is likely to be a more accurate representation of the true proportion of woody productivity in the canopy than the lower estimate based on average growth over the entire 19-year period. Because canopy status changes over time, measurements of growth closer in time to the canopy status assessments are more likely to be of trees in the same canopy status. In contrast, further back in time canopy status is increasingly likely to be different, which will decrease growth differences of trees classified as canopy versus understory. Thus, basing calculations only on the most recent 4 years leads to higher growth estimates for canopy versus understory trees, and a higher proportion of wood productivity in the canopy. In this study, we revealed the contributions of canopy and understory trees to carbon stocks and wood productivity, and at our knowledge we are the first study discussing this topic.

The differences in the size structure of canopy and understory trees observed in this study enable quantitative tests of previously presented theoretical models. Farrior et al. [31] developed a mechanistic model for the emergence of understory and canopy size distributions based on space-filling competition. Their model predicts that within individual patches, understory trees follow a power function and canopy trees follow an exponential distribution. Combining size distributions for patches of different ages under the assumption of a constant rate of patch disturbance results in predictions for the whole-forest size distributions. The overall predicted size distribution of smaller (mostly understory) trees is close to a power function (straight line on a log-log scale), whereas that of the larger (mostly canopy) trees is close to exponential (curving below a straight line in a J shape on a log-log scale). This is consistent with the size distribution observed here: if we take 25 cm as our empirically observed threshold diameter at which 50% of trees are in the canopy as a cutoff, the size distribution for smaller trees is close to a straight line on log-log scales, whereas for larger trees it is J-shaped (Fig 5c). Also consistent with Farrior et al. [31], the best-fit size distribution for canopy trees > 25 cm is the exponential distribution. The good fit of the Weibull to the whole-forest size distribution is also consistent with demographic equilibrium theory [34, 36, 49]. In contrast, the observed stand-level size distribution was inconsistent with metabolic theory, as the power function was a poor fit, and the best-fit power exponent was significantly different from -2.

In conducting the field work linking crowns to tagged stems, we observed a number of distinct strategies by which trees increased their access to light, strategies that would not have

been apparent from either the drone-acquired imagery or the ground field work alone. We observed that some individuals extend long branches beneath the crowns of other trees to reach a gap in which they produce a second set of leaves. From the orthomosaic image alone, these would appear to be two separate crowns belonging to different individuals, highlighting the importance of the field work. Other trees lean sharply such that their crowns are strongly displaced from the rooting points of their trunks. Many small trees emerge from underneath the crowns of larger trees, and a substantial number of trees with DBH less than 10 cm are exposed to direct sunlight. These observations corroborate other studies of the plasticity of crowns [50, 51] and light as a highly influential factor in forest dynamics [17, 36]. It is important to note that many trees that have crowns within the plot have their trunks outside of the plot, and vice versa. This causes misinterpretation among the relationships of forest inventory data and remote sensing. Essentially this is an edge effect problem, with more severe errors for smaller plots [52].

## Conclusion

The identification and field mapping of crowns seen in a high resolution orthomosaic revealed new patterns in the structure and dynamics of trees occupying different light environments in this Amazonian forest. In this study, we were able to determine canopy status of individual trees, and thereby quantify the proportion of trees in canopy and understory in relation to tree size, the contributions of canopy and understory trees to carbon stocks and wood productivity, and differences in stem growth and size distributions between canopy and understory trees. Less than half of the trees with DBH > 10 cm were in the canopy, but they were disproportionately larger trees and accounted for ~70% of carbon stocks and wood productivity. Diameter growth rates of canopy trees were unrelated to diameter, suggesting that the general increase in growth with diameter is due to greater light exposure for larger trees. The size distributions of understory and canopy trees were consistent with mechanistic models based on steady state emerging from local competition. Thus, this study demonstrates how the combination of high-resolution aerial imagery and ground-based field work has great potential to improve our understanding of the structure and dynamics of old-growth tropical forests having dense understories.

## Supporting information

**S1 Table. Number and proportion of trees in the canopy by 10 cm diameter class.**
(DOCX)

**S1 Fig. Mean diameter growth rates of individual trees in relation to their DBH and canopy status.** The dashed and dash-dot lines show the mean growth rates of 2.34 and 1.18 mm. $year^{-1}$ for the canopy and understory groups, respectively.
(TIF)

**S2 Fig. Probability to occupy the canopy status.** Canopy status in relation to DBH (a) and MAI (b) for individual trees (points), together with fitted logistic regressions (lines). The fitted lines cross 50% for at DBH of 23.5 cm and MAI of 2.2 mm.$year^{-1}$.
(TIF)

**S3 Fig. Exponential fit for canopy trees.** Observed size distribution of canopy trees (green circles) together with the exponential fit probability density functions (green line). The x-axis is the stem diameter class in cm, on a linear scale; the y-axis is the frequency of individuals per hectare per 1-cm size class.
(TIF)

**S4 Fig. Exponential, power and Weibull fits for all, understory and canopy trees.** All 1244 trees with DBH > 10 cm (black points), understory trees (gray points), and canopy trees (green circles), shown together with fit probability density functions (lines). Distributions are shown on linear (top) and log (bottom) scales. The vertical dashed black line indicates the minimum diameter (25 cm) for inclusion in the canopy tree fits. The Weibull distribution (Eq 6) had the best fit for all individuals combined; the exponential distribution (Eq 4) had the best fit for understory trees as well as for canopy trees with DBH $\geq$ 25 cm (solid lines). The other fits are shown by dashed and dotted lines. Whereas data are graphed here for 10-cm size classes for visualization purposes, fits were carried out using 1-cm size classes (parameter values in Table 2).
(TIF)

**S1 File. Shapefile of transect NS subplots.**
(ZIP)

**S2 File. Table containing forest inventory and crown delineation data.**
(CSV)

**S3 File. Metadata explaining the S2 File contents.**
(TXT)

**S4 File. Orthomosaic image resized to 50 cm spatial resolution.**
(TIF)

**S5 File. Shapefile of crown delineation polygons.**
(ZIP)

## Acknowledgments

We gratefully acknowledge logistical support by the staff of the INPA Laboratório de Manejo Florestal. We thank the project Monitoramento Ambiental por Satélite no Bioma Amazônia (MSA)—Subprojeto 7 for the ALS data. We also thank Camille Piponiot for helping with the manuscript submission. KC Cushman, Evan Gora, Teja Kattenborn and one anonymous reviewer provided helpful comments on earlier versions of this manuscript.

## Author Contributions

**Conceptualization:** Raquel Fernandes Araujo, Jeffrey Q. Chambers, Carlos Henrique Souza Celes.

**Data curation:** Adriano J. N. Lima, Niro Higuchi.

**Formal analysis:** Raquel Fernandes Araujo, Carlos Henrique Souza Celes, Helene C. Muller-Landau, Fabiano Emmert, Gabriel H. P. M. Ribeiro.

**Methodology:** Raquel Fernandes Araujo, Jeffrey Q. Chambers, Carlos Henrique Souza Celes, Helene C. Muller-Landau, Ana Paula Ferreira dos Santos, Fabiano Emmert, Gabriel H. P. M. Ribeiro, Bruno Oliva Gimenez, Niro Higuchi.

**Software:** Moacir A. A. Campos.

**Writing – original draft:** Raquel Fernandes Araujo.

**Writing – review & editing:** Raquel Fernandes Araujo, Jeffrey Q. Chambers, Carlos Henrique Souza Celes, Helene C. Muller-Landau, Ana Paula Ferreira dos Santos, Fabiano Emmert, Gabriel H. P. M. Ribeiro, Bruno Oliva Gimenez.

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
