## [Decision Letter · Decision Letter 0]

30 Jun 2020

PONE-D-20-08690

Forest dynamics and canopy structure from a high resolution remotely piloted aircraft imagery in the Central Amazon

PLOS ONE

Dear Dr. Araujo,

Thank you for submitting your manuscript to PLOS ONE. After careful consideration, we feel that it has merit but does not fully meet PLOS ONE’s publication criteria as it currently stands. Therefore, we invite you to submit a revised version of the manuscript that addresses the points raised during the review process.

We look forward to receiving your revised manuscript.

Kind regards,

Jana Müllerová, Ph.D

Academic Editor

PLOS ONE

Journal Requirements:

2. We note that Figure 1 in your submission contain [map/satellite] images which may be copyrighted. All PLOS content is published under the Creative Commons Attribution License (CC BY 4.0), which means that the manuscript, images, and Supporting Information files will be freely available online, and any third party is permitted to access, download, copy, distribute, and use these materials in any way, even commercially, with proper attribution. For these reasons, we cannot publish previously copyrighted maps or satellite images created using proprietary data, such as Google software (Google Maps, Street View, and Earth). For more information, see our copyright guidelines: http://journals.plos.org/plosone/s/licenses-and-copyright.

Additional Editor Comments (if provided):

The paper is interesting and research well structured. Therefore only minor revision is recommended. Please make sure you address all the issues raised by the reviewers, especially to define aims and research gaps you want to address in your study, add details on data collection and processing, and formulate clear conclusions.

Reviewers' comments:

Reviewer's Responses to Questions

**Comments to the Author**

1. Is the manuscript technically sound, and do the data support the conclusions?

Reviewer #1: Yes

Reviewer #2: Yes

2. Has the statistical analysis been performed appropriately and rigorously? 

Reviewer #1: Yes

Reviewer #2: Yes

3. Have the authors made all data underlying the findings in their manuscript fully available?

Reviewer #1: No

Reviewer #2: No

4. Is the manuscript presented in an intelligible fashion and written in standard English?

Reviewer #1: Yes

Reviewer #2: Yes

5. Review Comments to the Author

Reviewer #1: The study is exploring the usage of high resolution orthomosaics from UAV to identify the crowns within an Amazon forests but also authors are combining the data with data from ground measurements started in 1996. Altogether 1244 trees from 51 subplots (20x20) are investigated. The number of trees and also the temporal ground data are great basis for all analysis. The main idea is not to assess the performance of UAV for crown delineation but to use it as one of the data sources to come with conclusions. In general, I like the whole concept of the study, where authors are taking the UAV in account and they are aware of the shortcomings of it (not seeing understory trees). The main result is that 40% of trees are canopy trees but they are accounted for 70% of aboveground carbon stocks and wood productivity.

I have comments mainly for methods. I have not found flaws or main issues in introduction, results or discussion.

TITLE

I would rather use “Unmanned aerial vehicle” or “Unmanned aircraft system”. I understand that the RPAS is a most formal term but nowadays it is not used frequently within manuscripts. But if you decided to use this formal name correct the title: Forest dynamics and canopy structure from a high resolution remotely piloted aircraft system imagery in the Central Amazon.

DATA AVAILABILITY

You have stated that your data are fully available. Could you add it also to the article? Especially the way how to access them. This will give additional value to your paper in my opinion.

Suppl. Shp: I have downloaded “Shapefile_Crowns_WGS84_UTM20S” but I it was not possible to open it. It seems empty because the extent is 0,0,0,0.

METHODS

In general, you should add more details in the part dedicated to data collection and processing. It is not clear how did you georeferenced the data from UAV but also those on the ground. Or everything was in local coordinates? If you have used to georeferenced and scale your data based on Flytrex then you should add more info about this device and the reliability of it.

Secondly, the crown fitting with ground data is not very clear.

L146: Please add information for the overlap. Whether it is related to ground or crown.

L146-147: What is the accuracy of the Flytrex core 2.0. You should add it to the article.

L182: would be good to add also the number of trees that were considered for that analysis.

Reviewer #2: The presented study compares forest inventory data with tree crowns visible in high resolution UAV imagery. This study reveals that there is no trivial connection between forest structure from the ground and from the air, and that a combination of both perspectives can provide important insights. The manuscript thereby provide interest to a wide readership. The manuscript is generally well written and the methods seem sound.

General comments:

I recommend to streamline the introduction and define a proper research gap. After reading the introduction for the fist time I had problems to get a clear picture of the overall research aim. The research questions are very well defined but came a bit as a surprise to me. I guess you should drop some details and focus on a clear red thread (too much detail may distract from the overall red thread).

The results regarding the visibility in the imagery as a function of the diameter distributions are indeed interesting! The same applies for the growth rates. The manuscript as a whole will certainly benefit from clearer guidance on these aspects in the introduction.

Regarding the DBH and canopy trees relationship: Did you compare different species or at least genera? It is very likely that the different species show a very different relationship since they are likely to have different strategies. Some species may show a conservative growth and accumulate a lot of resources (e.g. high wood density). Other species may be more competitive and grow fast in height but not in DBH (they may aim to overtop neighbors). I was missing at least a discussion on this. A separation into species or genera (at least the dominant ones) may also allow to fit better models. I guess you would also see a very strong pattern in the species distribution as a function of canopy trees since shadow-tolerant species usually feature smaller heights.

I missed a conclusion section where you distil your main findings. This would also enable to create a nice frame for the manuscript and relate to your initial research questions.

I hope this review helps to further improve the manuscript. Best regards, Teja Kattenborn

Detailed comments:

Title: I think the title does not really resemble the content of your paper. I guess it would be better to include the aspect of comparing UAV with inventory data – this is what I understood the research gap and should, hence, be reflected in the title.

l.27 Crucial for what? To me the research gap is somewhere between the lines, but ideally the research gap should be explicitly formulated. Are you primarily interested in the forest structure and dynamics or in the methodology of deriving the latter?

l.31 The term ‚canopy trees‘ sounds odd to me, since every tree contributes to the canopy. This becomes even more clear when considering that a canopy can have multiple layers. At least you should clearly define what you mean with canopy trees before using such concept.

l.46 Limitted in terms of what? And does the number of studies matter or is it rather the missing knowledge that matters?

l.51 LiDAR does penetrate up to the ground. This should be reflected here.

l.63-72 I do not really understand how this paragraphs contributes to formulating the research gap. To me this is rather off-topic or distracting, respectively.

l.97 I cannot really follow the red line of your introduction. You start with tropical forest, then with remote sensing, then you jump again to forest structures and again to remote sensing. I recommend to streamline your introduction and address one topic after another. For instance along these lines: 1) Knowledge on the forest structuere in tropical forest remains scarce. There are different models and empirical findings. Inventory data alone is unlikely to reveal the structural diversity and causal processes. Remote sensing offers insights from another perspective. Combining Inventory data and remote sensing is key to address the above mentioned research gap.

l.104 There is no reference here. I guess this one suitable: Kattenborn, T., Lopatin, J., Förster, M., Braun, A. C., & Fassnacht, F. E. (2019). UAV data as alternative to field sampling to map woody invasive species based on combined Sentinel-1 and Sentinel-2 data. Remote Sensing of Environment, https://doi.org/10.1016/j.rse.2019.03.025

l.117 This section most importantly misses a motivation on why you chose this study area.

l.151-156 This description is a bit short in my opinion. To my experience from various similar works, linking UAV and in-situ data can be very challenging. Did you use a RTK-based GNSS? Based on what criteria did you assign a species to a stem coordinate? When looking at the RGB imagery with the red polygons it seems to me that there are some uncertainties with the delineation of the tree crowns. For instance polygon #1022, #1018 and #1035 seem to cover different tree species / individuals. How can you be sure that these interpretations are plausible? Consider to elaborate on your methods and discuss potential problems.

l.163 There might be gap constellations that enable a tree to receive direct sunlight for the major part of the day although these trees might not be fully visible from the bird perspective. From a process-based perspective such trees could be considered as ‘canopy trees’. You should consider to discuss such problems, since your method may not perfectly resemble the inherent process of the system.

Table 2 Maybe it would be worth to visualize the distributions vs your observations. I guess this would greatly help the reader to get an understanding of the data structure and the distributions.

l.294 This sentence may imply that you applied an automated crown delineation. Remote Sensing itself cannot detect trees, but an algorithm or interpreter can do. Consider to revise this sentence.

l.303 Here it appears that tree growth only depends on light. One would have to know the growth rate of small trees with sufficient access to light to make such a statement – at least I would state this more carefully. Especially considering that tropical forests are often rather limited by nutrients rather than by light. It could also be that the larger root system of bigger trees is indeed being the causal factor that drives higher growth rates. I do not want to state that your statement is entirely wrong – I just recommend to be more careful.

l.339 I do not really agree here. You indeed showed that there are some statistical patterns, but you did not present causal relationships (see comments above on species, nutrients, etc…).

l.343ff This connects to the issue I addressed above – given these aspects, how can you be sure that you delineated the trees appropriately? This should be discussed in my opinion.

l.361 I totally agree – maybe this aspect could be nicely paraphrased by saying that different perspectives, i.e. the ‘ground’ and the ‘bird perspective’, are needed to fully understand the complex structure of forests. In the light of your results it may also be worth to discuss that passive optical remote sensing is certainly limited in revealing the forest structure or its biomass. In this regard I was also missing reference to other studies that used more sophisticated sensors (e.g. LiDAR) to describe forest structures.

The data is not made available yet. There is no inventory data, no orthoimagery and the supplied shapefile is empty.

6. PLOS authors have the option to publish the peer review history of their article (what does this mean?). If published, this will include your full peer review and any attached files.

Reviewer #1: No

Reviewer #2: **Yes: **Teja Kattenborn

---

## [Author Response · Author response to Decision Letter 0]

14 Sep 2020

September 13, 2020

To:

Jana Müllerová, Academic Editor

PLOS ONE

With regards to our submission (PONE-D-20-08690), please find below an accounting of the changes we have made to address the helpful comments of the reviewers and editor. We have reformulated the introduction to better motivate the research questions, added details on data collection and processing, and revised the discussion including formulating a final conclusion paragraph. Note that we have changed the title, as suggested by both reviewers. We hope that the revised manuscript is now acceptable for publication in PLOS ONE. We thank the reviewers and editors for their contributions.

Sincerely,

Raquel Fernandes de Araujo

DETAILED RESPONSES IN ITALICS

Journal Requirements:

We note that Figure 1 in your submission contain [map/satellite] images which may be copyrighted.

R: Thank you for pointing this out. The images of Figure 1 are from US Geological Survey and they may be used and reproduced without copyright restriction. Following the USGS recommendation, we included the citation in the Figure 1 legend: “Landsat-8 (a, b) and SRTM (c) images courtesy of the U.S. Geological Survey”.

Reviewer #1

Title

I would rather use “Unmanned aerial vehicle” or “Unmanned aircraft system”. I understand that the RPAS is a most formal term but nowadays it is not used frequently within manuscripts. But if you decided to use this formal name correct the title: Forest dynamics and canopy structure from a high resolution remotely piloted aircraft system imagery in the Central Amazon.

R: We changed RPAS to drone in the title, as this is the more common popular usage these days. Within the manuscript itself, we use RPAS after defining it. We modified the title for: “Integrating high resolution drone imagery and forest inventory to distinguish canopy and understory trees and quantify their contributions to forest structure and dynamics”

Data availability

You have stated that your data are fully available. Could you add it also to the article? Especially the way how to access them. This will give additional value to your paper in my opinion.

Suppl. Shp: I have downloaded “Shapefile_Crowns_WGS84_UTM20S” but I it was not possible to open it. It seems empty because the extent is 0,0,0,0.

R: We uploaded a correct crown delineation shapefile. We also included the forest inventory data, the orthomosaic image resized to 50 cm (resized due to file size limit to upload in the PLOS ONE system), and the shapefile of subplots as supporting information files. We double-checked the files and all of them are opening in GIS software.

Methods

In general, you should add more details in the part dedicated to data collection and processing. It is not clear how did you georeferenced the data from UAV but also those on the ground. Or everything was in local coordinates? If you have used to georeferenced and scale your data based on Flytrex then you should add more info about this device and the reliability of it.

R: Thank you for pointing this out. As requested, we added more details on the data collection and processing, including an explanation of the georeferencing. Indeed, we installed the GPS track logger model Flytrex Core 2.0, that recorded the flight geographic coordinates. But we realized that we used Flytrex data only in another analysis not included in this paper.

We used an Airborne Laser Scanning (ALS) data to georeference the orthomosaic. There the text now reads:

“We processed photos using the photogrammetry software Agisoft Photoscan (https://www.agisoft.com, v.1.3.0), which aligned the photos using the Scale Invariant Feature Transformation (SIFT) algorithm and produced a point cloud model based on overlap among photos. (Because the camera was not integrated with the RPAS, GPS coordinates of the flight were not automatically assigned to photos). We georeferenced this model using as reference an Airborne Laser Scanning (ALS) dataset from the North-South Transect plot. We selected 15 control points evenly distributed across the flight area and extracted the XYZ coordinates (UTM – Universal Transverse Mercator projection Zone 20S, WGS84 horizontal datum). The control points were the center of crowns and palm trees, and the solar panels of two towers of the AmazonFACE project located in the plot, all well visible in both ALS data and photos. Georeferencing accuracy was assessed in terms of the Root Mean Square Error (RMSE) reported by the software.”

Secondly, the crown fitting with ground data is not very clear.

R: We revised the text to more fully explain these methods. The text now reads:

“Maps with the orthomosaic were printed for each 20 m x 20 m subplot and crowns visible in the image were identified in the field, with reference to the forest inventory data on individual tree DBH, species identity, and tag number. We started delineating the crowns of trees with the biggest trunk diameters, looking from the base to the top and identifying the whole boundary of each crown. We drew the tree crowns boundaries on the map with their respective tag numbers. We then delineated the crowns of the smaller trees, identifying the neighboring trees which had crowns surrounding the biggest crowns. This in situ delineation of crowns allowed us to perceive that in some cases branches of the same tree, but located in different heights and positions, had different colors in the orthomosaic, such that examination of the imagery alone would suggest they belong to different individuals. After the field work, all the crowns were vectorized in GIS, with the creation of polygons to represent each crown (Fig 2, S5 File).”

L146: Please add information for the overlap. Whether it is related to ground or crown.

R: As requested, we added information on the overlap:

“The minimum longitudinal and side overlap were 88% and 78%, respectively, in areas of peak canopy height on ridges; overlap was larger in areas with lower canopy height and in slope and valley areas.”

L146-147: What is the accuracy of the Flytrex core 2.0. You should add it to the article.

R: As explained above, we installed the GPS track logger model Flytrex Core 2.0, that recorded the flight geographic coordinates. But we realized that we used Flytrex data only in another analysis not included in this paper, so we removed all mention of Flytrex from the methods.

L182: would be good to add also the number of trees that were considered for that analysis.

R: Following the suggestion, we added “n = 484 and 696 trees” for canopy and understory trees, respectively.

Reviewer #2

General comments:

I recommend to streamline the introduction and define a proper research gap. After reading the introduction for the fist time I had problems to get a clear picture of the overall research aim. The research questions are very well defined but came a bit as a surprise to me. I guess you should drop some details and focus on a clear red thread (too much detail may distract from the overall red thread).

The results regarding the visibility in the imagery as a function of the diameter distributions are indeed interesting! The same applies for the growth rates. The manuscript as a whole will certainly benefit from clearer guidance on these aspects in the introduction.

R: Thank you for the constructive criticism. We have now completely rewritten the introduction to better motivate the research questions and clarify the contribution of this work.

Regarding the DBH and canopy trees relationship: Did you compare different species or at least genera? It is very likely that the different species show a very different relationship since they are likely to have different strategies. Some species may show a conservative growth and accumulate a lot of resources (e.g. high wood density). Other species may be more competitive and grow fast in height but not in DBH (they may aim to overtop neighbors). I was missing at least a discussion on this. A separation into species or genera (at least the dominant ones) may also allow to fit better models. I guess you would also see a very strong pattern in the species distribution as a function of canopy trees since shadow-tolerant species usually feature smaller heights.

R: We agree that including species or genus in the analysis would give more information about different strategies to reach light. However, because of the high tree diversity in this forest, we have limited samples sizes for individual species, and thus we chose not to pursue this analysis with this dataset. We hope to explore it with a larger dataset in a future study. 

I missed a conclusion section where you distil your main findings. This would also enable to create a nice frame for the manuscript and relate to your initial research questions.

R: As requested, we now included a conclusion section.

Detailed comments:

Title: I think the title does not really resemble the content of your paper. I guess it would be better to include the aspect of comparing UAV with inventory data – this is what I understood the research gap and should, hence, be reflected in the title.

R: Following the suggestion, we modified the title to: “Integrating high resolution drone imagery and forest inventory to distinguish canopy and understory and quantify their contributions to forest structure and dynamics”

l.27 Crucial for what? To me the research gap is somewhere between the lines, but ideally the research gap should be explicitly formulated. Are you primarily interested in the forest structure and dynamics or in the methodology of deriving the latter?

R: We reformulate this statement. There the text now reads:

“The integration of remote sensing and ground-based data enables this determination and measurements of how canopy and understory trees differ in structure and dynamics.”

l.31 The term ‚canopy trees‘ sounds odd to me, since every tree contributes to the canopy. This becomes even more clear when considering that a canopy can have multiple layers. At least you should clearly define what you mean with canopy trees before using such concept.

R: We agree that “canopy tree” has different meanings in different papers, which can be confusing. We considered a number of different options for wording (overstory vs. understory, exposed vs. nonexposed), but in the end it seemed to us that canopy and understory were the best terms to use. Our usage of these terms is consistent with a considerable prior literature, e.g., Bohlman 2015. We have now added parenthetical definition of what is meant by a canopy tree in the abstract at its first mention, and also stated the definition at greater length in the methods. 

l.46 Limitted in terms of what? And does the number of studies matter or is it rather the missing knowledge that matters?

R: We completely rewrote this paragraph, and this sentence no longer appears.

l.51 LiDAR does penetrate up to the ground. This should be reflected here.

R: We completely rewrote this paragraph, and this sentence no longer appears.

l.63-72 I do not really understand how this paragraphs contributes to formulating the research gap. To me this is rather off-topic or distracting, respectively.

R: Following your suggestion, we reformulated this paragraph.

l.97 I cannot really follow the red line of your introduction. You start with tropical forest, then with remote sensing, then you jump again to forest structures and again to remote sensing. I recommend to streamline your introduction and address one topic after another. For instance along these lines: 1) Knowledge on the forest structuere in tropical forest remains scarce. There are different models and empirical findings. Inventory data alone is unlikely to reveal the structural diversity and causal processes. Remote sensing offers insights from another perspective. Combining Inventory data and remote sensing is key to address the above mentioned research gap.

R: Thank you for the constructive criticism. As noted above, we have now completely rewritten the introduction to better motivate the research questions and clarify the contribution of this work, very much in line with these suggestions. 

l.104 There is no reference here. I guess this one suitable: Kattenborn, T., Lopatin, J., Förster, M., Braun, A. C., & Fassnacht, F. E. (2019). UAV data as alternative to field sampling to map woody invasive species based on combined Sentinel-1 and Sentinel-2 data. Remote Sensing of Environment, https://doi.org/10.1016/j.rse.2019.03.025

R: We included the reference, as suggested.

l.117 This section most importantly misses a motivation on why you chose this study area.

R: As requested, we included the motivation

“The availability of ~two decades of bi-annual tree censuses and the proximity from the road (Fig 1) enabled the accomplishment of the present study in this plot.”

l.151-156 This description is a bit short in my opinion. To my experience from various similar works, linking UAV and in-situ data can be very challenging. Did you use a RTK-based GNSS? Based on what criteria did you assign a species to a stem coordinate? When looking at the RGB imagery with the red polygons it seems to me that there are some uncertainties with the delineation of the tree crowns. For instance polygon #1022, #1018 and #1035 seem to cover different tree species / individuals. How can you be sure that these interpretations are plausible? Consider to elaborate on your methods and discuss potential problems.

R: We added more detail on data collection and processing, explaining better how we georeferenced the orthomosaic and linked crowns to ground data.

“We processed photos using the photogrammetry software Agisoft Photoscan (https://www.agisoft.com, v.1.3.0), which aligned the photos using the Scale Invariant Feature Transformation (SIFT) algorithm and produced a point cloud model based on overlap among photos. (Because the camera was not integrated with the RPAS, GPS coordinates of the flight were not automatically assigned to photos). We georeferenced this model using as reference an Airborne Laser Scanning (ALS) dataset from the North-South Transect plot. We selected 15 control points evenly distributed across the flight area and extracted the XYZ coordinates (UTM – Universal Transverse Mercator projection Zone 20S, WGS84 horizontal datum). The control points were the center of crowns and palm trees, and the solar panels of two towers of the AmazonFACE project located in the plot, all well visible in both ALS data and photos. Georeferencing accuracy was assessed in terms of the Root Mean Square Error (RMSE) reported by the software.”

“Maps with the orthomosaic were printed for each 20 m x 20 m subplot and crowns visible in the image were identified in the field, with reference to the forest inventory data on individual tree DBH, species identity, and tag number. We started delineating the crowns of trees with the biggest trunk diameters, looking from the base to the top and identifying the whole boundary of each crown. We drew the tree crowns boundaries on the map with their respective tag numbers. We then delineated the crowns of the smaller trees, identifying the neighboring trees which had crowns surrounding the biggest crowns. This in situ delineation of crowns allowed us to perceive that in some cases branches of the same tree, but located in different heights and positions, had different colors in the orthomosaic, such that examination of the imagery alone would suggest they belong to different individuals. After the field work, all the crowns were vectorized in GIS, with the creation of polygons to represent each crown (Fig 2, S5 File).”

l.163 There might be gap constellations that enable a tree to receive direct sunlight for the major part of the day although these trees might not be fully visible from the bird perspective. From a process-based perspective such trees could be considered as ‘canopy trees’. You should consider to discuss such problems, since your method may not perfectly resemble the inherent process of the system.

R: We agree that some of the trees we classified as “understory” trees may nonetheless receive direct lateral light for part of the day. We now explicitly acknowledge these cases and other limitations of our definitions when we state it in the methods. In full, this section now reads “For the purposes of this study, canopy trees were defined as those directly exposed to overhead light; other trees were classified as understory trees (as in Bohlman, 2015). Specifically, trees were classified as being in the canopy if their crowns were totally or partially visible in the orthomosaic, and had a visible crown diameter greater than ~ 1.5 m. We note that some trees classified as canopy trees under this definition may have most of their crowns shaded and only a small part of the crown in direct light, and that small trees can be classed as canopy trees if they do not have larger trees above them. We also note that some trees classified as understory trees under this definition may received direct lateral light for part of the day.”

Table 2 Maybe it would be worth to visualize the distributions vs your observations. I guess this would greatly help the reader to get an understanding of the data structure and the distributions.

R: This is an excellent suggestion. We included a new figure in the supplementary material (S4 Figure) showing fits for all models.

l.294 This sentence may imply that you applied an automated crown delineation. Remote Sensing itself cannot detect trees, but an algorithm or interpreter can do. Consider to revise this sentence.

R: Thank you for pointing this out. We reformulated this sentence:

“Canopy trees represented less than half of trees with DBH > 10 cm…”

l.303 Here it appears that tree growth only depends on light. One would have to know the growth rate of small trees with sufficient access to light to make such a statement – at least I would state this more carefully. Especially considering that tropical forests are often rather limited by nutrients rather than by light. It could also be that the larger root system of bigger trees is indeed being the causal factor that drives higher growth rates. I do not want to state that your statement is entirely wrong – I just recommend to be more careful.

R: As suggested, we modified to “Light availability is one factor that directly influences tree growth in tropical forests.”

l.339 I do not really agree here. You indeed showed that there are some statistical patterns, but you did not present causal relationships (see comments above on species, nutrients, etc…).

R: We recognize that our wording was potentially misleading. We modified this sentence to:

“In conducting the field work linking crowns to tagged stems, we observed a number of distinct strategies by which trees increased their access to light, strategies that would not have been apparent from either the drone-acquired imagery or the ground field work alone.”

l.343ff This connects to the issue I addressed above – given these aspects, how can you be sure that you delineated the trees appropriately? This should be discussed in my opinion.

R: We delineated crowns at field, we after vectorized in GIS, and returned to the field to check remaining doubts in the crown delineation.

l.361 I totally agree – maybe this aspect could be nicely paraphrased by saying that different perspectives, i.e. the ‘ground’ and the ‘bird perspective’, are needed to fully understand the complex structure of forests. In the light of your results it may also be worth to discuss that passive optical remote sensing is certainly limited in revealing the forest structure or its biomass. In this regard I was also missing reference to other studies that used more sophisticated sensors (e.g. LiDAR) to describe forest structures.

R: Indeed, the penetrability of airborne laser scanning is important to measure tree height and describe forest structure and dynamics. In this study we referenced studies using LiDAR in the introduction (reference numbers 1,30). We chose not discussing them because the LiDAR it’s not the focus of our study.

The data is not made available yet. There is no inventory data, no orthoimagery and the supplied shapefile is empty.

R: We included the forest inventory data, the orthomosaic image resized to 50 cm (resized due to file size limit to upload in the PLOS ONE system), the shapefiles of subplots and crown delineation as supporting information files. We double-checked the files and all of them are opening in GIS software.

---

## [Decision Letter · Decision Letter 1]

7 Oct 2020

PONE-D-20-08690R1

Integrating high resolution drone imagery and forest inventory to distinguish canopy and understory trees and quantify their contributions to forest structure and dynamics

PLOS ONE

Dear Dr. Araujo,

Thank you for submitting your manuscript to PLOS ONE. After careful consideration, we feel that it has merit but does not fully meet PLOS ONE’s publication criteria as it currently stands. Therefore, we invite you to submit a revised version of the manuscript that addresses the points raised during the review process.

We look forward to receiving your revised manuscript.

Kind regards,

Jana Müllerová, Ph.D

Academic Editor

PLOS ONE

Additional Editor Comments (if provided):

Dear authors. I am happy to see that your manuscript improved very much, now it is clear and informative. Only very few issues remain. After you addressed those, your paper will be ready for publication. Good job indeed!

Reviewers' comments:

Reviewer's Responses to Questions

**Comments to the Author**

1. If the authors have adequately addressed your comments raised in a previous round of review and you feel that this manuscript is now acceptable for publication, you may indicate that here to bypass the “Comments to the Author” section, enter your conflict of interest statement in the “Confidential to Editor” section, and submit your "Accept" recommendation.

Reviewer #2: All comments have been addressed

2. Is the manuscript technically sound, and do the data support the conclusions?

Reviewer #2: Yes

3. Has the statistical analysis been performed appropriately and rigorously? 

Reviewer #2: Yes

4. Have the authors made all data underlying the findings in their manuscript fully available?

Reviewer #2: Yes

5. Is the manuscript presented in an intelligible fashion and written in standard English?

Reviewer #2: Yes

6. Review Comments to the Author

Reviewer #2: The revised manuscript clearly improved in clarity and reading flow. All of my previous comments were addressed and misunderstandings were clarified. The manuscript reads nicely and the results are now more elegantly highlighted in the added conclusion section.

Besides some details (see below), the only further suggestion I have relates to a comment of the previous review: I suggested to study the species as a factor of canopy proportions, DBH and growth rates. I undersand, that your data may not enable to do such analysis. Nevertheless, I would (briefly! - maybe 1-2 sentences) discuss that looking deeper into species/genus-specific relationships may shed additional light on the forest structural heterogeneity. Especially, considering that some species are by design shadow tolerant (there is no need to reach the canopy to thrive), whereas other species (competitive species) may only succeed on the long run, if the eventually reach the canopy. There may, for example, be only a few species with exceptional high canopy heights containing the majority of timber. I would consider to add such thoughts to 1) explain the scatter in your results and 2) provide an outlook for future work.

Good job!

Teja Kattenborn

Minor Details:

l.141 Remove full stop before braket.

l.149 Consider to state that you generated the products using Agisoft.

l.310 Full stop after and not before the bracket.

Fig.5 In the text, you primarily refer to DBH, whereas in the figures you use ‘Diamter’ as axis label. Consider to also use DBH in the figures. Diameter is rather unspecific (and could, for example, also refer to crown diameter).

7. PLOS authors have the option to publish the peer review history of their article (what does this mean?). If published, this will include your full peer review and any attached files.

Reviewer #2: **Yes: **Teja Kattenborn

---

## [Author Response · Author response to Decision Letter 1]

20 Oct 2020

October 20, 2020

To:

Jana Müllerová, Academic Editor

PLOS ONE

With regards to our submission (PONE-D-20-08690R1), please find below the changes we have made to address the helpful comments of the reviewer. We have added the discussion of investigating the proportion of trees in the canopy per species/genus and addressed the minor suggestions. We thank the reviewers and editors for their contributions.

Sincerely,

Raquel Fernandes de Araujo

DETAILED RESPONSES IN ITALICS

Reviewer #2

Besides some details (see below), the only further suggestion I have relates to a comment of the previous review: I suggested to study the species as a factor of canopy proportions, DBH and growth rates. I undersand, that your data may not enable to do such analysis. Nevertheless, I would (briefly! - maybe 1-2 sentences) discuss that looking deeper into species/genus-specific relationships may shed additional light on the forest structural heterogeneity. Especially, considering that some species are by design shadow tolerant (there is no need to reach the canopy to thrive), whereas other species (competitive species) may only succeed on the long run, if the eventually reach the canopy. There may, for example, be only a few species with exceptional high canopy heights containing the majority of timber. I would consider to add such thoughts to 1) explain the scatter in your results and 2) provide an outlook for future work.

R: As suggested, we added these thoughts in the second paragraph of discussion section. There the text now reads:

“Investigating the proportion of trees in the canopy per species or genus should bring more information on forest structure heterogeneity. At BCI, the proportion of gap-dependent species was higher in the canopy than the understory and increased with tree size, while the proportion of shade-tolerant species was smaller in the canopy and decreased with tree size [27]. Future studies should evaluate if the PPA can explain differences in the numbers and sizes of canopy trees among forests such as those between our site and BCI and explore differences in species and functional group compositions between canopy and understory.”

Minor Details:

l.141 Remove full stop before bracket.

R: Thank you for pointing this out. We removed the full stop before the bracket.

l.149 Consider to state that you generated the products using Agisoft.

R: We reformulate this statement. There the text now reads:

“We then generated a 3D point cloud, a digital elevation model (DEM) and a 2 cm spatial resolution orthomosaic (e.g., Fig 2, S4 File) using Agisoft Photoscan.”

l.310 Full stop after and not before the bracket.

R: We replaced the full stop after the bracket (this is in the l.242).

Fig.5 In the text, you primarily refer to DBH, whereas in the figures you use ‘Diameter’ as axis label. Consider to also use DBH in the figures. Diameter is rather unspecific (and could, for example, also refer to crown diameter).

R: Following the suggestion, we replaced the x axis labels of all figures with “DBH (cm)”.

---

## [Editor Report · Decision Letter 2]

16 Nov 2020

Integrating high resolution drone imagery and forest inventory to distinguish canopy and understory trees and quantify their contributions to forest structure and dynamics

PONE-D-20-08690R2

Dear Dr. Araujo,

We’re pleased to inform you that your manuscript has been judged scientifically suitable for publication and will be formally accepted for publication once it meets all outstanding technical requirements.

Kind regards,

Jana Müllerová, Ph.D

Academic Editor

PLOS ONE

Additional Editor Comments (optional):

Congrats to your paper, it is now ready for publication. Regards
---

## [Editor Report · Acceptance letter]

1 Dec 2020

PONE-D-20-08690R2 

Integrating high resolution drone imagery and forest inventory to distinguish canopy and understory trees and quantify their contributions to forest structure and dynamics 

Dear Dr. Araujo:

I'm pleased to inform you that your manuscript has been deemed suitable for publication in PLOS ONE. Congratulations! Your manuscript is now with our production department. 

Kind regards, 

on behalf of

Dr. Jana Müllerová 

Academic Editor

PLOS ONE